# Hessian-free Optimization for Learning
# Deep Multidimensional Recurrent Neural Networks

**Minhyung Cho**     **Chandra Shekhar Dhir**     **Jaehyung Lee**
Applied Research Korea, Gracenote Inc.
{mhyung.cho,shekhardhir}@gmail.com     jaehyung.lee@kaist.ac.kr

## Abstract

Multidimensional recurrent neural networks (MDRNNs) have shown a remarkable performance in the area of speech and handwriting recognition. The performance of an MDRNN is improved by further increasing its depth, and the difficulty of learning the deeper network is overcome by using Hessian-free (HF) optimization. Given that connectionist temporal classification (CTC) is utilized as an objective of learning an MDRNN for sequence labeling, the non-convexity of CTC poses a problem when applying HF to the network. As a solution, a convex approximation of CTC is formulated and its relationship with the EM algorithm and the Fisher information matrix is discussed. An MDRNN up to a depth of 15 layers is successfully trained using HF, resulting in an improved performance for sequence labeling.

## 1   Introduction

Multidimensional recurrent neural networks (MDRNNs) constitute an efficient architecture for building a multidimensional context into recurrent neural networks [1]. End-to-end training of MDRNNs in conjunction with connectionist temporal classification (CTC) has been shown to achieve a state-of-the-art performance in on/off-line handwriting and speech recognition [2, 3, 4].

In previous approaches, the performance of MDRNNs having a depth of up to five layers, which is limited as compared to the recent progress in feedforward networks [5], was demonstrated. The effectiveness of MDRNNs deeper than five layers has thus far been unknown.

Training a deep architecture has always been a challenging topic in machine learning. A notable breakthrough was achieved when deep feedforward neural networks were initialized using layer-wise pre-training [6]. Recently, approaches have been proposed in which supervision is added to intermediate layers to train deep networks [5, 7]. To the best of our knowledge, no such pre-training or bootstrapping method has been developed for MDRNNs.

Alternatively, Hesssian-free (HF) optimization is an appealing approach to training deep neural networks because of its ability to overcome pathological curvature of the objective function [8]. Furthermore, it can be applied to any connectionist model provided that its objective function is differentiable. The recent success of HF for deep feedforward and recurrent neural networks [8, 9] supports its application to MDRNNs.

In this paper, we claim that an MDRNN can benefit from a deeper architecture, and the application of second order optimization such as HF allows its successful learning. First, we offer details of the development of HF optimization for MDRNNs. Then, to apply HF optimization for sequence labeling tasks, we address the problem of the non-convexity of CTC, and formulate a convex approximation. In addition, its relationship with the EM algorithm and the Fisher information matrix is discussed. Experimental results for offline handwriting and phoneme recognition show that an MDRNN with HF optimization performs better as the depth of the network increases up to 15 layers.

## 2   Multidimensional recurrent neural networks

MDRNNs constitute a generalization of RNNs to process multidimensional data by replacing the single recurrent connection with as many connections as the dimensions of the data [1]. The network can access the contextual information from $2^N$ directions, allowing a collective decision to be made based on rich context information. To enhance its ability to exploit context information, long short-term memory (LSTM) [10] cells are usually utilized as hidden units. In addition, stacking MDRNNs to construct deeper networks further improves the performance as the depth increases, achieving the state-of-the-art performance in phoneme recognition [4]. For sequence labeling, CTC is applied as a loss function of the MDRNN. The important advantage of using CTC is that no pre-segmented sequences are required, and the entire transcription of the input sample is sufficient.

### 2.1   Learning MDRNNs

A $d$-dimensional MDRNN with $M$ inputs and $K$ outputs is regarded as a mapping from an input sequence $\mathbf{x} \in \mathbb{R}^{M \times T_1 \times \cdots \times T_d}$ to an output sequence $\mathbf{a} \in (\mathbb{R}^K)^T$ of length $T$, where the input data for $M$ input neurons are given by the vectorization of $d$-dimensional data, and $T_1, \ldots, T_d$ is the length of the sequence in each dimension. All learnable weights and biases are concatenated to obtain a parameter vector $\theta \in \mathbb{R}^N$. In the learning phase with fixed training data, the MDRNN is formalized as a mapping $\mathcal{N} : \mathbb{R}^N \to (\mathbb{R}^K)^T$ from the parameters $\theta$ to the output sequence $\mathbf{a}$, i.e., $\mathbf{a} = \mathcal{N}(\theta)$. The scalar loss function is defined over the output sequence as $\mathcal{L} : (\mathbb{R}^K)^T \to \mathbb{R}$. Learning an MDRNN is viewed as an optimization of the objective $\mathcal{L}(\mathcal{N}(\theta)) = \mathcal{L} \circ \mathcal{N}(\theta)$ with respect to $\theta$.

The Jacobian $J_{\mathcal{F}}$ of a function $\mathcal{F} : \mathbb{R}^m \to \mathbb{R}^n$ is the $n \times m$ matrix where each element is a partial derivative of an element of output with respect to an element of input. The Hessian $H_{\mathcal{F}}$ of a scalar function $\mathcal{F} : \mathbb{R}^m \to \mathbb{R}$ is the $m \times m$ matrix of second-order partial derivatives of the output with respect to its inputs. Throughout this paper, a vector sequence is denoted by boldface $\mathbf{a}$, a vector at time $t$ in $\mathbf{a}$ is denoted by $a^t$, and the $k$-th element of $a^t$ is denoted by $a_k^t$.

## 3   Hessian-free optimization for MDRNNs

The application of HF optimization to an MDRNN is straightforward if the matching loss function [11] for its output layer is adopted. However, this is not the case for CTC, which is necessarily adopted for sequence labeling. Before developing an appropriate approximation to CTC that is compatible with HF optimization, we discuss two considerations related to the approximation. The first is obtaining a quadratic approximation of the loss function, and the second is the efficient calculation of the matrix-vector product used at each iteration of the conjugate gradient (CG) method.

HF optimization minimizes an objective by constructing a local quadratic approximation for the objective function and minimizing the approximate function instead of the original one. The loss function $\mathcal{L}(\theta)$ needs to be approximated at each point $\theta_n$ of the $n$-th iteration:

$$Q_n(\theta) = \mathcal{L}(\theta_n) + \nabla_\theta \mathcal{L}|_{\theta_n}^\top \delta_n + \frac{1}{2} \delta_n^\top G \delta_n, \tag{1}$$

where $\delta_n = \theta - \theta_n$ is the search direction, i.e., the parameters of the optimization, and $G$ is a local approximation to the curvature of $\mathcal{L}(\theta)$ at $\theta_n$, which is typically obtained by the generalized Gauss-Newton (GGN) matrix as an approximation of the Hessian.

HF optimization uses the CG method in a subroutine to minimize the quadratic objective above for utilizing the complete curvature information and achieving computational efficiency. CG requires the computation of $Gv$ for an arbitrary vector $v$, but not the explicit evaluation of $G$. For neural networks, an efficient way to compute $Gv$ was proposed in [11], extending the study in [12]. In section 3.2, we provide the details of the efficient computation of $Gv$ for MDRNNs.

### 3.1   Quadratic approximation of loss function

The Hessian matrix, $H_{\mathcal{L} \circ \mathcal{N}}$, of the objective $\mathcal{L}(\mathcal{N}(\theta))$ is written as

$$H_{\mathcal{L} \circ \mathcal{N}} = J_{\mathcal{N}}^\top H_{\mathcal{L}} J_{\mathcal{N}} + \sum_{i=1}^{KT} [J_{\mathcal{L}}]_i H_{[\mathcal{N}]_i}, \tag{2}$$

where $J_\mathcal{N} \in \mathbb{R}^{KT \times N}$, $H_\mathcal{L} \in \mathbb{R}^{KT \times KT}$, and $[q]_i$ denotes the $i$-th component of the vector $q$. An indefinite Hessian matrix is problematic for second-order optimization, because it defines an unbounded local quadratic approximation [13]. For nonlinear systems, the Hessian is not necessarily positive semidefinite, and thus, the GGN matrix is used as an approximation of the Hessian [11, 8]. The GGN matrix is obtained by ignoring the second term in Eq. (2), as given by

$$G_{\mathcal{L} \circ \mathcal{N}} = J_\mathcal{N}^\top H_\mathcal{L} J_\mathcal{N}. \tag{3}$$

The sufficient condition for the GGN approximation to be exact is that the network makes a perfect prediction for every given sample, that is, $J_\mathcal{L} = 0$, or $[\mathcal{N}]_i$ stays in the linear region for all $i$, that is, $H_{[\mathcal{N}]_i} = 0$.

$G_{\mathcal{L} \circ \mathcal{N}}$ has less rank than $KT$ and is positive semidefinite provided that $H_\mathcal{L}$ is. Thus, $\mathcal{L}$ is chosen to be a convex function so that $H_\mathcal{L}$ is positive semidefinite. In principle, it is best to define $\mathcal{L}$ and $\mathcal{N}$ such that $\mathcal{L}$ performs as much of the computation as possible, with the positive semidefiniteness of $H_\mathcal{L}$ as a minimum requirement [13]. In practice, a nonlinear output layer together with its matching loss function [11], such as the softmax function with cross-entropy loss, is widely used.

## 3.2 Computation of matrix-vector product for MDRNN

The product of an arbitrary vector $v$ by the GGN matrix, $Gv = J_\mathcal{N}^\top H_\mathcal{L} J_\mathcal{N} v$, amounts to the sequential multiplication of $v$ by three matrices. First, the product $J_\mathcal{N} v$ is a Jacobian times vector and is therefore equal to the directional derivative of $\mathcal{N}(\theta)$ along the direction of $v$. Thus, $J_\mathcal{N} v$ can be written using a differential operator $J_\mathcal{N} v = \mathcal{R}_v(\mathcal{N}(\theta))$ [12] and the properties of the operator can be utilized for efficient computation. Because an MDRNN is a composition of differentiable components, the computation of $\mathcal{R}_v(\mathcal{N}(\theta))$ throughout the whole network can be accomplished by repeatedly applying the sum, product, and chain rules starting from the input layer. The detailed derivation of the $\mathcal{R}$ operator to LSTM, normally used as a hidden unit in MDRNNs, is provided in appendix A.

Next, the multiplication of $J_\mathcal{N} v$ by $H_\mathcal{L}$ can be performed by direct computation. The dimension of $H_\mathcal{L}$ could at first appear problematic, since the dimension of the output vector used by the loss function $\mathcal{L}$ can be as high as $KT$, in particular, if CTC is adopted as an objective for the MDRNN. If the loss function can be expressed as the sum of individual loss functions with a domain restricted in time, the computation can be reduced significantly. For example, with the commonly used cross-entropy loss function, the $KT \times KT$ matrix $H_\mathcal{L}$ can be transformed into a block diagonal matrix with $T$ blocks of a $K \times K$ Hessian matrix. Let $H_{\mathcal{L},t}$ be the $t$-th block in $H_\mathcal{L}$. Then, the GGN matrix can be written as

$$G_{\mathcal{L} \circ \mathcal{N}} = \sum_t J_{\mathcal{N}_t}^\top H_{\mathcal{L},t} J_{\mathcal{N}_t}, \tag{4}$$

where $J_{\mathcal{N}_t}$ is the Jacobian of the network at time $t$.

Finally, the multiplication of a vector $u = H_\mathcal{L} J_\mathcal{N} v$ by the matrix $J_\mathcal{N}^\top$ is calculated using the back-propagation through time algorithm by propagating $u$ instead of the error at the output layer.

## 4 Convex approximation of CTC for application to HF optimization

Connectioninst temporal classification (CTC) [14] provides an objective function of learning an MDRNN for sequence labeling. In this section, we derive a convex approximation of CTC inspired by the GGN approximation according to the following steps. First, the non-convex part of the original objective is separated out by reformulating the softmax part. Next, the remaining convex part is approximated without altering its Hessian, making it well matched to the non-convex part. Finally, the convex approximation is obtained by reuniting the convex and non-convex parts.

### 4.1 Connectionist temporal classification

CTC is formulated as the mapping from an output sequence of the recurrent network, $\mathbf{a} \in (\mathbb{R}^K)^T$, to a scalar loss. The output activations at time $t$ are normalized using the softmax function

$$y_k^t = \frac{\exp(a_k^t)}{\sum_{k'} \exp(a_{k'}^t)}, \tag{5}$$

where $y_k^t$ is the probability of label $k$ given $\mathbf{a}$ at time $t$.

The conditional probability of the path $\pi$ is calculated by the multiplication of the label probabilities at each timestep, as given by

$$p(\pi|\mathbf{a}) = \prod_{t=1}^{T} y_{\pi_t}^t, \tag{6}$$

where $\pi_t$ is the label observed at time $t$ along the path $\pi$. The path $\pi$ of length $T$ is mapped to a label sequence of length $M \leq T$ by an operator $\mathcal{B}$, which removes the repeated labels and then the blanks. Several mutually exclusive paths can map to the same label sequence. Let $S$ be a set containing every possible sequence mapped by $\mathcal{B}$, that is, $S = \{s | s \in \mathcal{B}(\pi) \text{ for some } \pi\}$ is the image of $\mathcal{B}$, and let $|S|$ denote the cardinality of the set.

The conditional probability of a label sequence $\mathbf{l}$ is given by

$$p(\mathbf{l}|\mathbf{a}) = \sum_{\pi \in \mathcal{B}^{-1}(\mathbf{l})} p(\pi|\mathbf{a}), \tag{7}$$

which is the sum of probabilities of all the paths mapped to a label sequence $\mathbf{l}$ by $\mathcal{B}$.

The cross-entropy loss assigns a negative log probability to the correct answer. Given a target sequence $\mathbf{z}$, the loss function of CTC for the sample is written as

$$\mathcal{L}(\mathbf{a}) = -\log p(\mathbf{z}|\mathbf{a}). \tag{8}$$

From the description above, CTC is composed of the sum of the product of softmax components. The function $-\log(y_k^t)$, corresponding to the softmax with cross-entropy loss, is convex [11]. Therefore, $y_k^t$ is log-concave. Whereas log-concavity is closed under multiplication, the sum of log-concave functions is not log-concave in general [15]. As a result, the CTC objective is not convex in general because it contains the sum of softmax components in Eq. (7).

## 4.2 Reformulation of CTC objective function

We reformulate the CTC objective Eq. (8) to separate out the terms that are responsible for the non-convexity of the function. By reformulation, the softmax function is defined over the categorical label sequences.

By substituting Eq. (5) into Eq. (6), it follows that

$$p(\pi|\mathbf{a}) = \frac{\exp(b_\pi)}{\sum_{\pi' \in \text{all}} \exp(b_{\pi'})}, \tag{9}$$

where $b_\pi = \sum_t a_{\pi_t}^t$. By substituting Eq. (9) into Eq. (7) and setting $\mathbf{l} = \mathbf{z}$, $p(\mathbf{z}|\mathbf{a})$ can be re-written as

$$p(\mathbf{z}|\mathbf{a}) = \frac{\sum_{\pi \in \mathcal{B}^{-1}(\mathbf{z})} \exp(b_\pi)}{\sum_{\pi \in \text{all}} \exp(b_\pi)} = \frac{\exp(f_\mathbf{z})}{\sum_{\mathbf{z}' \in S} \exp(f_{\mathbf{z}'})}, \tag{10}$$

where $S$ is the set of every possible label sequence and $f_\mathbf{z} = \log\left(\sum_{\pi \in \mathcal{B}^{-1}(\mathbf{z})} \exp(b_\pi)\right)$ is the *log-sum-exp* function[1], which is proportional to the probability of observing the label sequence $\mathbf{z}$ among all the other label sequences.

With the reformulation above, the CTC objective can be regarded as the cross-entropy loss with the softmax output, which is defined over all the possible label sequences. Because the cross-entropy loss function matches the softmax output layer [11], the CTC objective is convex, except the part that computes $f_\mathbf{z}$ for each of the label sequences. At this point, an obvious candidate for the convex approximation of CTC is the GGN matrix separating the convex and non-convex parts.

Let the non-convex part be $\mathcal{N}_c$ and the convex part be $\mathcal{L}_c$. The mapping $\mathcal{N}_c : (\mathbb{R}^K)^T \to \mathbb{R}^{|S|}$ is defined by

$$\mathcal{N}_c(\mathbf{a}) = F = [f_{\mathbf{z}_1}, \ldots, f_{\mathbf{z}_{|S|}}]^\top, \tag{11}$$

where $f_{\mathbf{z}}$ is given above, and $|S|$ is the number of all the possible label sequences. For given $F$ as above, the mapping $\mathcal{L}_c : \mathbb{R}^{|S|} \to \mathbb{R}$ is defined by

$$\mathcal{L}_c(F) = -\log \frac{\exp(f_{\mathbf{z}})}{\sum_{\mathbf{z}' \in S} \exp(f_{\mathbf{z}'})} = -f_{\mathbf{z}} + \log \left( \sum_{\mathbf{z}' \in S} \exp(f_{\mathbf{z}'}) \right), \tag{12}$$

where $\mathbf{z}$ is the label sequence corresponding to $\mathbf{a}$. The final reformulation for the loss function of CTC is given by

$$\mathcal{L}(\mathbf{a}) = \mathcal{L}_c \circ \mathcal{N}_c(\mathbf{a}). \tag{13}$$

### 4.3 Convex approximation of CTC loss function

The GGN approximation of Eq. (13) immediately gives a convex approximation of the Hessian for CTC as $G_{\mathcal{L}_c \circ \mathcal{N}_c} = J_{\mathcal{N}_c}^\top H_{\mathcal{L}_c} J_{\mathcal{N}_c}$. Although $H_{\mathcal{L}_c}$ has the form of a diagonal matrix plus a rank-1 matrix, i.e., $\mathrm{diag}(Y) - YY^\top$, the dimension of $H_{\mathcal{L}_c}$ is $|S| \times |S|$, where $|S|$ becomes exponentially large as the length of the sequence increases. This makes the practical calculation of $H_{\mathcal{L}_c}$ difficult.

On the other hand, removing the linear team $-f_{\mathbf{z}}$ from $\mathcal{L}_c(F)$ in Eq. (12) does not alter its Hessian. The resulting formula is $\mathcal{L}_p(F) = \log \left( \sum_{\mathbf{z}' \in S} \exp(f_{\mathbf{z}'}) \right)$. The GGN matrices of $\mathcal{L} = \mathcal{L}_c \circ \mathcal{N}_c$ and $\mathcal{M} = \mathcal{L}_p \circ \mathcal{N}_c$ are the same, i.e., $G_{\mathcal{L}_c \circ \mathcal{N}_c} = G_{\mathcal{L}_p \circ \mathcal{N}_c}$. Therefore, their Hessian matrices are approximations of each other. The condition that the two Hessian matrices, $H_{\mathcal{L}}$ and $H_{\mathcal{M}}$, converges to the same matrix is discussed below.

Interestingly, $\mathcal{M}$ is given as a compact formula $\mathcal{M}(\mathbf{a}) = \mathcal{L}_p \circ \mathcal{N}_c(\mathbf{a}) = \sum_t \log \sum_k \exp(a_k^t)$, where $a_k^t$ is the output unit $k$ at time $t$. Its Hessian $H_{\mathcal{M}}$ can be directly computed, resulting in a block diagonal matrix. Each block is restricted in time, and the $t$-th block is given by

$$H_{\mathcal{M},t} = \mathrm{diag}(Y^t) - Y^t {Y^t}^\top, \tag{14}$$

where $Y^t = [y_1^t, \ldots, y_K^t]^\top$ and $y_k^t$ is given in Eq. (5). Because the Hessian of each block is positive semidefinite, $H_{\mathcal{M}}$ is positive semidefinite. A convex approximation of the Hessian of an MDRNN using the CTC objective can be obtained by substituting $H_{\mathcal{M}}$ for $H_{\mathcal{L}}$ in Eq. (3). Note that the resulting matrix is block diagonal and Eq. (4) can be utilized for efficient computation.

Our derivation can be summarized as follows:

1. $H_{\mathcal{L}} = H_{\mathcal{L}_c \circ \mathcal{N}_c}$ is not positive semidefinite.
2. $G_{\mathcal{L}_c \circ \mathcal{N}_c} = G_{\mathcal{L}_p \circ \mathcal{N}_c}$ is positive semidefinite, but not computationally tractable.
3. $H_{\mathcal{L}_p \circ \mathcal{N}_c}$ is positive semidefinite and computationally tractable.

### 4.4 Sufficient condition for the proposed approximation to be exact

From Eq. (2), the condition $H_{\mathcal{L}_c \circ \mathcal{N}_c} = H_{\mathcal{L}_p \circ \mathcal{N}_c}$ holds if and only if $\sum_{i=1}^{KT} [J_{\mathcal{L}_c}]_i H_{[\mathcal{N}_c]_i} = \sum_{i=1}^{KT} [J_{\mathcal{L}_p}]_i H_{[\mathcal{N}_c]_i}$. Since $J_{\mathcal{L}_c} \neq J_{\mathcal{L}_p}$ in general, we consider only the case of $H_{[\mathcal{N}_c]_i} = 0$ for all $i$, which corresponds to the case where $\mathcal{N}_c$ is a linear mapping.

$[\mathcal{N}_c]_i$ contains a *log-sum-exp* function mapping from paths to a label sequence. Let $\mathbf{l}$ be the label sequence corresponding to $[\mathcal{N}_c]_i$; then, $[\mathcal{N}_c]_i = f_{\mathbf{l}}(\ldots, b_\pi, \ldots)$ for $\pi \in \mathcal{B}^{-1}(\mathbf{l})$. If the probability of one path $\pi'$ is sufficiently large to ignore all the other paths, that is, $\exp(b_{\pi'}) \gg \exp(b_\pi)$ for $\pi \in \{\mathcal{B}^{-1}(\mathbf{l}) \backslash \pi'\}$, it follows that $f_{\mathbf{l}}(\ldots, b_{\pi'}, \ldots) = b_{\pi'}$. This is a linear mapping, which results in $H_{[\mathcal{N}_c]_i} = 0$.

In conclusion, the condition $H_{\mathcal{L}_c \circ \mathcal{N}_c} = H_{\mathcal{L}_p \circ \mathcal{N}_c}$ holds if one dominant path $\pi \in \mathcal{B}^{-1}(\mathbf{l})$ exists such that $f_{\mathbf{l}}(\ldots, b_\pi, \ldots) = b_\pi$ for each label sequence $\mathbf{l}$.

### 4.5 Derivation of the proposed approximation from the Fisher information matrix

The identity of the GGN and the Fisher information matrix [16] has been shown for the network using the softmax with cross-entropy loss [17, 18]. Thus, it follows that the GGN matrix of Eq. (13) is identical to the Fisher information matrix. Now, we show that the proposed matrix in Eq. (14)

is derived from the Fisher information matrix under the condition given in section 4.4. The Fisher information matrix of an MDRNN using CTC is written as

$$F = \mathbb{E}_{\mathbf{x}} \left[ J_{\mathcal{N}}^{\top} \mathbb{E}_{\mathbf{l} \sim p(\mathbf{l}|\mathbf{a})} \left[ \left( \frac{\partial \log p(\mathbf{l}|\mathbf{a})}{\partial \mathbf{a}} \right)^{\top} \left( \frac{\partial \log p(\mathbf{l}|\mathbf{a})}{\partial \mathbf{a}} \right) \right] J_{\mathcal{N}} \right], \tag{15}$$

where $\mathbf{a} = \mathbf{a}(\mathbf{x}, \theta)$ is the $KT$-dimensional output of the network $\mathcal{N}$. CTC assumes output probabilities at each timestep to be independent of those at other timesteps [1], and therefore, its Fisher information matrix is given as the sum of every timestep. It follows that

$$F = \mathbb{E}_{\mathbf{x}} \left[ \sum_{t} J_{\mathcal{N}_t}^{\top} \mathbb{E}_{\mathbf{l} \sim p(\mathbf{l}|\mathbf{a})} \left[ \left( \frac{\partial \log p(\mathbf{l}|\mathbf{a})}{\partial a^t} \right)^{\top} \left( \frac{\partial \log p(\mathbf{l}|\mathbf{a})}{\partial a^t} \right) \right] J_{\mathcal{N}_t} \right]. \tag{16}$$

Under the condition in section 4.4, the Fisher information matrix is given by

$$F = \mathbb{E}_{\mathbf{x}} \left[ \sum_{t} J_{\mathcal{N}_t}^{\top} (\mathrm{diag}(Y^t) - Y^t {Y^t}^{\top}) J_{\mathcal{N}_t} \right], \tag{17}$$

which is the same form as Eqs. (4) and (14) combined. See appendix B for the detailed derivation.

### 4.6 EM interpretation of the proposed approximation

The goal of the Expectation-Maximization (EM) algorithm is to find the maximum likelihood solution for models having latent variables [19]. Given an input sequence $\mathbf{x}$, and its corresponding target label sequence $\mathbf{z}$, the log likelihood of $\mathbf{z}$ is given by $\log p(\mathbf{z}|\mathbf{x}, \theta) = \log \sum_{\pi \in \mathcal{B}^{-1}(\mathbf{z})} p(\pi|\mathbf{x}, \theta)$, where $\theta$ represents the model parameters. For each observation $\mathbf{x}$, we have a corresponding latent variable $q$ which is a 1-of-$k$ binary vector where $k$ is the number of all the paths mapped to $\mathbf{z}$. The log likelihood can be written in terms of $q$ as $\log p(\mathbf{z}, q|\mathbf{x}, \theta) = \sum_{\pi \in \mathcal{B}^{-1}(\mathbf{z})} q_{\pi|\mathbf{x},\mathbf{z}} \log p(\pi|\mathbf{x}, \theta)$. The EM algorithm starts with an initial parameter $\hat{\theta}$, and repeats the following process until convergence.

Expectation step calculates: $\gamma_{\pi|\mathbf{x},\mathbf{z}} = \frac{p(\pi|\mathbf{x},\hat{\theta})}{\sum_{\pi \in \mathcal{B}^{-1}(\mathbf{z})} p(\pi|\mathbf{x},\hat{\theta})}$.

Maximization step updates: $\hat{\theta} = \mathrm{argmax}_{\theta} \mathcal{Q}(\theta)$, where $\mathcal{Q}(\theta) = \sum_{\pi \in \mathcal{B}^{-1}(\mathbf{z})} \gamma_{\pi|\mathbf{x},\mathbf{z}} \log p(\pi|\mathbf{x}, \theta)$.

In the context of CTC and RNN, $p(\pi|\mathbf{x}, \theta)$ is given as $p(\pi|\mathbf{a}(\mathbf{x}, \theta))$ as in Eq. (6), where $\mathbf{a}(\mathbf{x}, \theta)$ is the $KT$-dimensional output of the neural network. Taking the second-order derivative of $\log p(\pi|\mathbf{a})$ with respect to $a^t$ gives $\mathrm{diag}(Y^t) - Y^t {Y^t}^{\top}$, with $Y^t$ as in Eq. (14). Because this term is independent of $\pi$ and $\sum_{\pi \in \mathcal{B}^{-1}(\mathbf{z})} \gamma_{\pi|\mathbf{x},\mathbf{z}} = 1$, the Hessian of $\mathcal{Q}$ with respect to $a^t$ is given by

$$H_{\mathcal{Q},t} = \mathrm{diag}(Y^t) - Y^t {Y^t}^{\top}, \tag{18}$$

which is the same as the convex approximation in Eq. (14).

## 5  Experiments

In this section, we present the experimental results for two different sequence labeling tasks, offline handwriting recognition and phoneme recognition. The performance of Hessian-free optimization for MDRNNs with the proposed matrix is compared with that of stochastic gradient descent (SGD) optimization on the same settings.

### 5.1  Database and preprocessing

The IFN/ENIT Database [20] is a database of handwritten Arabic words, which consists of 32,492 images. The entire dataset has five subsets ($a$, $b$, $c$, $d$, $e$). The 25,955 images corresponding to the subsets ($b - e$) were used for training. The validation set consisted of 3,269 images corresponding to the first half of the sorted list in alphabetical order (ae07_001.tif − ai54_028.tif) in set $a$. The remaining images in set $a$, amounting to 3,268, were used for the test. The intensity of pixels was centered and scaled using the mean and standard deviation calculated from the training set.

The TIMIT corpus [21] is a benchmark database for evaluating speech recognition performance. The standard training, validation, and core datasets were used. Each set contains 3,696 sentences, 400 sentences, and 192 sentences, respectively. A mel spectrum with 26 coefficients was used as a feature vector with a pre-emphasis filter, 25 ms window size, and 10 ms shift size. Each input feature was centered and scaled using the mean and standard deviation of the training set.

## 5.2 Experimental setup

For handwriting recognition, the basic architecture was adopted from that proposed in [3]. Deeper networks were constructed by replacing the top layer with more layers. The number of LSTM cells in the augmented layer was chosen such that the total number of weights between the different networks was similar. The detailed architectures are described in Table 1, together with the results.

For phoneme recognition, the deep bidirectional LSTM and CTC in [4] was adopted as the basic architecture. In addition, the memory cell block [10], in which the cells share the gates, was applied for efficient information sharing. Each LSTM block was constrained to have 10 memory cells.

According to the results, using a large value of bias for input/output gates is beneficial for training deep MDRNNs. A possible explanation is that the activation of neurons is exponentially decayed by input/output gates during the propagation. Thus, setting large bias values for these gates may facilitate the transmission of information through many layers at the beginning of the learning. For this reason, the biases of the input and output gates were initialized to 2, whereas those of the forget gates and memory cells were initialized to 0. All the other weight parameters of the MDRNN were initialized randomly from a uniform distribution in the range $[-0.1, 0.1]$.

The label error rate was used as the metric for performance evaluation, together with the average loss of CTC in Eq. (8). It is defined by the edit distance, which sums the total number of insertions, deletions, and substitutions required to match two given sequences. The final performance, shown in Tables 1 and 2, was evaluated using the weight parameters that gave the best label error rate on the validation set. To map output probabilities to a label sequence, best path decoding [1] was used for handwriting recognition and beam search decoding [4, 22] with a beam width of 100 was used for phoneme recognition. For phoneme recognition, 61 phoneme labels were used during training and decoding, and then, mapped to 39 classes for calculating the phoneme error rate (PER) [4, 23].

For phoneme recognition, the regularization method suggested in [24] was used. We applied Gaussian weight noise of standard deviation $\sigma = \{0.03, 0.04, 0.05\}$ together with L2 regularization of strength $0.001$. The network was first trained without noise, and then, it was initialized to the weights that gave the lowest CTC loss on the validation set. Then, the network was retrained with Gaussian weight noise [4]. Table 2 presents the best result for different values of $\sigma$.

### 5.2.1 Parameters

For HF optimization, we followed the basic setup described in [8], but different parameters were utilized. Tikhonov damping was used together with Levenberg-Marquardt heuristics. The value of the damping parameter $\lambda$ was initialized to $0.1$, and adjusted according to the reduction ratio $\rho$ (multiplied by 0.9 if $\rho > 0.75$, divided by 0.9 if $\rho < 0.25$, and unchanged otherwise). The initial search direction for each run of CG was set to the CG direction found by the previous HF optimization iteration decayed by 0.7. To ensure that CG followed the descent direction, we continued to perform a minimum 5 and maximum 30 of additional CG iterations after it found the first descent direction. We terminated CG at iteration $i$ before reaching the maximum iteration if the following condition was satisfied: $(\phi(x_i) - \phi(x_{i-5}))/\phi(x_i) < 0.005$ , where $\phi$ is the quadratic objective of CG without offset. The training data were divided into 100 and 50 mini-batches for the handwriting and phoneme recognition experiments, respectively, and used for both the gradient and matrix-vector product calculation. The learning was stopped if any of two criteria did not improve for 20 epochs and 10 epochs in handwriting and phoneme recognition, respectively.

For SGD optimization, the learning rate $\epsilon$ was chosen from $\{10^{-4}, 10^{-5}, 10^{-6}\}$, and the momentum $\mu$ from $\{0.9, 0.95, 0.99\}$. For handwriting recognition, the best performance obtained using all the possible combinations of parameters is presented in Table 1. For phoneme recognition, the best parameters out of nine candidates for each network were selected after training without weight noise based on the CTC loss. Additionally, the backpropagated error in LSTM layer was clipped to remain

in the range $[-1, 1]$ for stable learning [25]. The learning was stopped after 1000 epochs had been processed, and the final performance was evaluated using the weight parameters that showed the best label error rate on the validation set. It should be noted that in order to guarantee the convergence, we selected a conservative criterion as compared to the study where the network converged after 85 epochs in handwriting recognition [3] and after 55-150 epochs in phoneme recognition [4].

## 5.3 Results

Table 1 presents the label error rate on the test set for handwriting recognition. In all cases, the networks trained using HF optimization outperformed those using SGD. The advantage of using HF is more pronounced as the depth increases. The improvements resulting from the deeper architecture can be seen with the error rate dropping from 6.1% to 4.5% as the depth increases from 3 to 13.

Table 2 shows the phoneme error rate (PER) on the core set for phoneme recognition. The improved performance according to the depth can be observed for both optimization methods. The best PER for HF optimization is 18.54% at 15 layers and that for SGD is 18.46% at 10 layers, which are comparable to that reported in [4], where the reported results are a PER of 18.6% from a network with 3 layers having 3.8 million weights and a PER of 18.4% from a network with 5 layers having 6.8 million weights. The benefit of a deeper network is obvious in terms of the number of weight parameters, although this is not intended to be a definitive performance comparison because of the different preprocessing. The advantage of HF optimization is not prominent in the result of the experiments using the TIMIT database. One explanation is that the networks tend to overfit to a relatively small number of the training data samples, which removes the advantage of using advanced optimization techniques.

Table 1: Experimental results for Arabic offline handwriting recognition. The label error rate is presented with the different network depths. $A^B$ denotes a stack of $B$ layers having $A$ hidden LSTM cells in each layer. "Epochs" is the number of epochs required by the network using HF optimization so that the stopping criteria are fulfilled. $\epsilon$ is the learning rate and $\mu$ is the momentum.

| NETWORKS | DEPTH | WEIGHTS | HF (%) | EPOCHS | SGD (%) | $\{\epsilon, \mu\}$ |
|---|---|---|---|---|---|---|
| 2-10-50 | 3 | 159,369 | 6.10 | 77 | 9.57 | $\{10^{-4}, 0.9\}$ |
| 2-10-21$^3$ | 5 | 157,681 | 5.85 | 90 | 9.19 | $\{10^{-5}, 0.99\}$ |
| 2-10-14$^6$ | 8 | 154,209 | 4.98 | 140 | 9.67 | $\{10^{-4}, 0.95\}$ |
| 2-10-12$^8$ | 10 | 154,153 | 4.95 | 109 | 9.25 | $\{10^{-4}, 0.95\}$ |
| 2-10-10$^{11}$ | 13 | 150,169 | 4.50 | 84 | 10.63 | $\{10^{-4}, 0.9\}$ |
| 2-10-9$^{13}$ | 15 | 145,417 | 5.69 | 84 | 12.29 | $\{10^{-5}, 0.99\}$ |

Table 2: Experimental results for phoneme recognition using the TIMIT corpus. PER is presented with the different MDRNN architectures (depth $\times$ block $\times$ cell/block). $\sigma$ is the standard deviation of Gaussian weight noise. The remaining parameters are the same as in Table 1.

| NETWORKS | WEIGHTS | HF (%) | EPOCHS | $\{\sigma\}$ | SGD (%) | $\{\epsilon, \mu, \sigma\}$ |
|---|---|---|---|---|---|---|
| $3 \times 20 \times 10$ | 771,542 | 20.14 | 22 | $\{0.03\}$ | 20.96 | $\{10^{-5}, 0.99, 0.05\}$ |
| $5 \times 15 \times 10$ | 795,752 | 19.18 | 30 | $\{0.05\}$ | 20.82 | $\{10^{-4}, 0.9, 0.04\}$ |
| $8 \times 11 \times 10$ | 720,826 | 19.09 | 29 | $\{0.05\}$ | 19.68 | $\{10^{-4}, 0.9, 0.04\}$ |
| $10 \times 10 \times 10$ | 755,822 | 18.79 | 60 | $\{0.04\}$ | 18.46 | $\{10^{-5}, 0.95, 0.04\}$ |
| $13 \times 9 \times 10$ | 806,588 | 18.59 | 93 | $\{0.05\}$ | 18.49 | $\{10^{-5}, 0.95, 0.04\}$ |
| $15 \times 8 \times 10$ | 741,230 | 18.54 | 50 | $\{0.04\}$ | 19.09 | $\{10^{-5}, 0.95, 0.03\}$ |
| $3 \times 250 \times 1^\dagger$ | 3.8M | | | | 18.6 | $\{10^{-4}, 0.9, 0.075\}$ |
| $5 \times 250 \times 1^\dagger$ | 6.8M | | | | 18.4 | $\{10^{-4}, 0.9, 0.075\}$ |

† The results were reported by Graves in 2013 [4].

## 6 Conclusion

Hessian-free optimization as an approach for successful learning of deep MDRNNs, in conjunction with CTC, was presented. To apply HF optimization to CTC, a convex approximation of its objective function was explored. In experiments, improvements in performance were seen as the depth of the network increased for both HF and SGD. HF optimization showed a significantly better performance for handwriting recognition than did SGD, and a comparable performance for speech recognition.

## Footnotes

[1] $f(x_1, \ldots, x_n) = \log(e^{x_1} + \cdots + e^{x_n})$ is the *log-sum-exp* function defined on $\mathbb{R}^n$

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
