[Supplementary Material]

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

We follow the version of LSTM in [4]. The forward pass of LSTM is given by

$$
\begin{aligned}
i_t &= \sigma(W_{xi}x_t + W_{hi}h_{t-1} + W_{ci}c_{t-1} + b_i), \\
f_t &= \sigma(W_{xf}x_t + W_{hf}h_{t-1} + W_{cf}c_{t-1} + b_f), \\
c_t &= f_t \cdot c_{t-1} + i_t \cdot \tanh(W_{xc}x_t + W_{hc}h_{t-1} + b_c), \\
o_t &= \sigma(W_{xo}x_t + W_{ho}h_{t-1}W_{co}c_t + b_o), \\
h_t &= o_t \cdot \tanh(c_t),
\end{aligned}
$$

where $\cdot$ denotes the element-wise vector product, $\sigma$ is the logistic sigmoid function, $x$, $h$, and $c$ are the input, hidden, and cell activation vector, respectively, and $i$, $o$, and $f$ are the input, output, and forget gates, respectively. All the gates and cells are the same size as the hidden vector $h$.

Applying the $\mathcal{R}$ operator to the above equations gives

$$
\begin{aligned}
\mathcal{R}_v(i_t) &= \sigma'(W_{xi}x_t + W_{hi}h_{t-1} + W_{ci}c_{t-1} + b_i) \\
&\quad \cdot (V_{xi}x_t + V_{hi}h_{t-1} + V_{ci}c_{t-1} + V_i + W_{hi}\mathcal{R}_v(h_{t-1}) + W_{ci}\mathcal{R}_v(c_{t-1})), \\
\mathcal{R}_v(f_t) &= \sigma'(W_{xf}x_t + W_{hf}h_{t-1} + W_{cf}c_{t-1} + b_f) \\
&\quad \cdot (V_{xf}x_t + V_{hf}h_{t-1} + V_{cf}c_{t-1} + V_f + W_{hf}\mathcal{R}_v(h_{t-1}) + W_{cf}\mathcal{R}_v(c_{t-1})), \\
\mathcal{R}_v(c_t) &= \mathcal{R}_v(f_t) \cdot c_{t-1} + f_t \cdot \mathcal{R}_v(c_{t-1}) + \mathcal{R}_v(i_t) \cdot \tanh(W_{xc}x_t + W_{hc}h_{t-1} + b_c) \\
&\quad + i_t \cdot \tanh'(W_{xc}x_t + W_{hc}h_{t-1} + b_c) \cdot (V_{xc}x_t + V_{hc}h_{t-1} + V_c + W_{hc}\mathcal{R}_v(h_{t-1})), \\
\mathcal{R}_v(o_t) &= \sigma'(W_{xo}x_t + W_{ho}h_{t-1} + W_{co}c_t + b_o), \\
&\quad \cdot (V_{xo}x_t + V_{ho}h_{t-1} + V_{co}c_t + V_o + W_{ho}\mathcal{R}_v(h_{t-1}) + W_{co}\mathcal{R}_v(c_t)), \\
\mathcal{R}_v(h_t) &= \mathcal{R}_v(o_t) \cdot \tanh(c_t) + o_t \cdot \tanh'(c_t) \cdot \mathcal{R}_v(c_t),
\end{aligned}
$$

where $V_{ij}$ and $V_i$ are taken from $v$ at the same point of $W_{ij}$ and $b_i$ in $\theta$, respectively. Note that $\theta$ and $v$ have the same dimension.

# B Detailed derivation of the proposed approximation from the Fisher information matrix

The derivative of the negative log probability of Eq. (7) is given by

$$
-\frac{\partial \log p(\mathbf{l}|\mathbf{a})}{\partial a_k^t} = y_k^t - \frac{1}{p(\mathbf{l}|\mathbf{a})} \sum_{s \in lab(\mathbf{l},k)} \alpha_t(s)\beta_t(s). \tag{19}
$$

where $\alpha_t(s)$ and $\beta_t(s)$ denote forward and backward variables, respectively, and $lab(\mathbf{l}, k) = \{u|\mathbf{l}_u = k\}$ is the set of positions, where label $k$ occurs in $\mathbf{l}$ [1, 3]. For compact notation, let $Y^t$ denote a column matrix containing $y_k^t$ as its $k$-th element, and let $V^t$ denote a column matrix containing $v_k^t = \frac{1}{p(\mathbf{l}|\mathbf{a})} \sum_{s \in lab(\mathbf{l},k)} \alpha_t(s)\beta_t(s)$ as its $k$-th element.

The Fisher information matrix [16] is defined by

$$
F = \mathbb{E}_{\mathbf{x}}\left[\mathbb{E}_{\mathbf{l}\sim p(\mathbf{l}|\mathbf{x})}\left[\left(\frac{\partial \log p(\mathbf{l}|\mathbf{x},\theta)}{\partial\theta}\right)^{\top}\left(\frac{\partial \log p(\mathbf{l}|\mathbf{x},\theta)}{\partial\theta}\right)\right]\right]. \tag{20}
$$

The Fisher information matrix of an MDRNN using CTC is written as

$$
F = \mathbb{E}_{\mathbf{x}}\left[\mathbb{E}_{\mathbf{l}\sim p(\mathbf{l}|\mathbf{x})}\left[\left(\frac{\partial \log p(\mathbf{l}|\mathbf{a})}{\partial\mathbf{a}}J_{\mathcal{N}}\right)^{\top}\left(\frac{\partial \log p(\mathbf{l}|\mathbf{a})}{\partial\mathbf{a}}J_{\mathcal{N}}\right)\right]\right] \tag{21}
$$

$$
= \mathbb{E}_{\mathbf{x}}\left[J_{\mathcal{N}}^{\top}\mathbb{E}_{\mathbf{l}\sim p(\mathbf{l}|\mathbf{a})}\left[\left(\frac{\partial \log p(\mathbf{l}|\mathbf{a})}{\partial\mathbf{a}}\right)^{\top}\left(\frac{\partial \log p(\mathbf{l}|\mathbf{a})}{\partial\mathbf{a}}\right)\right]J_{\mathcal{N}}\right], \tag{22}
$$

where $\mathbf{a} = \mathbf{a}(\mathbf{x}, \theta)$ is the $KT$-dimensional output of the network $\mathcal{N}$. The final step follows from that $J_{\mathcal{N}}$ is independent of $\mathbf{l}$.

CTC assumes the output probabilities at each timestep to be independent of those at other timesteps [1], and therefore, its Fisher information matrix is given as the sum of every timestep. It follows that

$$F = \mathbb{E}_{\mathbf{x}} \left[ \sum_t J_{\mathcal{N}_t}^\top \mathbb{E}_{\mathbf{l} \sim p(\mathbf{l}|\mathbf{a})} \left[ \left( \frac{\partial \log p(\mathbf{l}|\mathbf{a})}{\partial a^t} \right)^\top \left( \frac{\partial \log p(\mathbf{l}|\mathbf{a})}{\partial a^t} \right) \right] J_{\mathcal{N}_t} \right] \tag{23}$$

$$= \mathbb{E}_{\mathbf{x}} \left[ \sum_t J_{\mathcal{N}_t}^\top \mathbb{E}_{\mathbf{l} \sim p(\mathbf{l}|\mathbf{a})} \left[ \left( Y^t - V^t \right) \left( Y^t - V^t \right)^\top \right] J_{\mathcal{N}_t} \right] \tag{24}$$

$$= \mathbb{E}_{\mathbf{x}} \left[ \sum_t J_{\mathcal{N}_t}^\top \left( Y^t Y^{t\top} - Y^t \mathbb{E}_{\mathbf{l}} \left[ V^t \right]^\top - \mathbb{E}_{\mathbf{l}} \left[ V^t \right] Y^{t\top} + \mathbb{E}_{\mathbf{l}} \left[ V^t V^{t\top} \right] \right) J_{\mathcal{N}_t} \right], \tag{25}$$

where $Y^t$ and $V^t$ are defined above.

$\mathbb{E}_{\mathbf{l}}[v_k^t]$ is given by

$$\mathbb{E}_{\mathbf{l}}[v_k^t] = \mathbb{E}_{\mathbf{l} \sim p(\mathbf{l}|\mathbf{a})} \left[ \frac{1}{p(\mathbf{l}|\mathbf{a})} \sum_{s \in lab(\mathbf{l},k)} \alpha_t(s)\beta_t(s) \right] \tag{26}$$

$$= \sum_{\mathbf{l}} \sum_{s \in lab(\mathbf{l},k)} \alpha_t(s)\beta_t(s) \tag{27}$$

$$= y_k^t. \tag{28}$$

$\mathbb{E}_{\mathbf{l}}[v_i^t v_j^t]$ is given by

$$\mathbb{E}_{\mathbf{l}}[v_i^t v_j^t] = \mathbb{E}_{\mathbf{l} \sim p(\mathbf{l}|\mathbf{a})} \left[ \frac{1}{p(\mathbf{l}|\mathbf{a})^2} \sum_{s \in lab(\mathbf{l},i)} \alpha_t(s)\beta_t(s) \sum_{s \in lab(\mathbf{l},j)} \alpha_t(s)\beta_t(s) \right]. \tag{29}$$

Unfortunately Eq. (29) cannot be analytically calculated in general. We apply the sufficient condition for the proposed approximation to be exact in section 4.4. By the assumption of one dominant path in a label sequence, $\mathbb{E}_{\mathbf{l}}[v_i^t v_j^t] = 0$ for $i \neq j$. If the dominant path visits $i$ at time $t$, $\sum_{s \in lab(\mathbf{l},i)} \alpha_t(s)\beta_t(s) = p(\mathbf{l}|\mathbf{a})$. Otherwise $\sum_{s \in lab(\mathbf{l},i)} \alpha_t(s)\beta_t(s) = 0$. Under this condition, Eq. (29) can be written as

$$\mathbb{E}_{\mathbf{l}}[v_i^t v_j^t] = \delta_{ij} \sum_{\mathbf{l}} \sum_{s \in lab(\mathbf{l},i)} \alpha_t(s)\beta_t(s) \tag{30}$$

$$= \delta_{ij} y_i^t, \tag{31}$$

where $\delta_{ij}$ is the Kronecker delta. Substituting $\mathbb{E}_{\mathbf{l}}[V^t] = Y^t$ and $\mathbb{E}_{\mathbf{l}}[V^t V^{t\top}] = \mathrm{diag}(Y^t)$ into Eq. (25) gives

$$F = \mathbb{E}_{\mathbf{x}} \left[ \sum_t J_{\mathcal{N}_t}^\top (\mathrm{diag}(Y^t) - Y^t Y^{t\top}) J_{\mathcal{N}_t} \right], \tag{32}$$

which is the same form as Eqs. (4) and (14) combined.