[Reviews · NeurIPS 2015]

Submitted by Assigned_Reviewer_1

This paper proposes using HF for learning a MDRNN with a CTC loss on top. The contributions of this work are to show that a loss that is non-convex (CTC) can still be used within Hessian Free by means of decomposing the loss using similar tricks than Gauss Newton matrices (which are applied already in HF methods for NNs). The authors show an elegant connection between their effort to make CTC suitable for HF, and an EM procedure.

The authors of this paper showed an effective way to train a very complex model (MDRNN), with a non-linear cost (CTC) using second order information (HF). The main theoretical contribution is to make CTC compatible with HF by proposing a GGN that is PSD, a requirement needed for CG within HF. Their contribution goes beyond that as they show interesting links with EM, and provide empirical evidence that the method is working.

I, however, have a few concerns regarding the experiments presented in this work:

-As the authors discuss, TIMIT is mostly an overfitting game and not so much and underfitting / optimization game. What was the training cost achieved by both methods?

-There are other baselines that may be more practical (i.e. less implementation details that may go wrong), such as Adagrad, RMSProp, or Adam. Could the authors provide such baselines? -It would be useful to add some prior art in the tables for comparison.

As a final note, there are quite a lot of grammatical and minor mistakes in the text. I'd suggest a deep pass over those before publishing this work.
Summary: Despite some problems regarding grammar, this paper shows large benefits of training a very complex model (MDRNN) with a quite complicated loss (CTC) using Hessian Free optimization. They compensate some technical difficulties (non-convexity of CTC) quite elegantly, and show rather strong evidence that their framework is working on two different tasks.

Submitted by Assigned_Reviewer_2

The paper proposes a Hessian-free optimization method to train deep Multi

Dimensional RNNs in the context of connectionist temporal classification

using a local quadratic

approximation of the objective function The approximation is computed using a low-rank updates while maintaining positive-semidefinitivness. Experimental comparisons are provided for the Arabic hand-writing recognition benchmark and on the TIMIT phoneme recognition corpus. Significant quality improvements are observed for the former.

The technical concent of this paper is sound and the quality of writing is high. The paper discusses relations with the Fisher information matrix and with expectation maximization, but the most straightforward

L-BFGS baseline is not discussed. It would have been helpful if the paper had provided at least experimental comparisons with the L-BFGS baseline, not just with SGD. Other problem is that the networks studied in the paper have relatively low parameter count. It is unclear how the quality improvements and runtime scales for networks with higher parameter count.
Summary: The paper proposes a Hessian-free optimization method to train deep multi dimensional RNNs in the context of connectionist temporal classification

using a local quadratic

approximation of the objective function. This is an interesting work but important practical and theoretical comparisons with other optimization methods like L-BFGS are missing.

Submitted by Assigned_Reviewer_3

This paper discusses how the authors applied Hessian Free optimization to MDRNs.

For me the deal-breaker was that the experimental comparison was very insufficient.

The authors decided to train the baseline SGD to convergence (1000 iterations) even though that almost never gives the best results.

They say "Note that in order to guarantee the convergence, we selected a conservative criteria compared to the reference".

I can think of no other reason to do this other than to artificially handicap the baseline and to have some good-sounding results to report.

Also there is no discussion of relative speed of the two methods, no convergence plots, and no objective functions reported, so the reader really has no idea what is going on.

Summary: The idea to apply Hessian Free optimization to that type of network potentially makes sense, there is weakness in the experiments.

Submitted by Assigned_Reviewer_4

A possible drawback is that the paper focuses a lot on details of the original work [11], and it is not that easy to judge the novelty of the authors' contributions related to [11].
Summary: Light review of Paper #570: "Hessian-Free Optimization For Learning Deep Multidimensional Recurrent Neural Networks"

The paper presents a sound idea to improve Hessian-free optimization for multi-dimensional recurrent neural networks combined with connectionist temporal classification for sequence labeling tasks. Overall, it is a clearly written paper with a convincing evaluation.

Submitted by Assigned_Reviewer_5

Table 1 looks convincing. But vanilla SGD is of course only one of many options which could be applied to this kind of optimization. For example, the sum-of-functions optimizer of Sohl-Dickstein et al. (2014) also uses second-order information and could presumably be applied without modification.

Regarding Table 2: As you mention, one explanation of the poor performance of HF might be overfitting. This could easily be checked by looking at the performance on the training data, so why not include it in the results?

The definition of S (page 4) is not clear to me, why does B map to a set and not a label sequence?
Summary: The paper's main contribution seems to be to make Hessian-free optimization of the CTC objective practical for MDRNNs. Based on a coarse reading, it is difficult for me to tell (a) what makes MDRNNs particularly difficult and why we couldn't just use the algorithm of Martens & Sutskever (2011), and (b) which aspects of Section 3 are new and which aspects are just a review of previous results.

Author Feedback
Author rebuttal: We would like to thank the reviewers (referred to as R) for their time and effort, and appreciate their constructive comments. Below, we attempt to address their comments to the best of our ability.

R1:
1) "... TIMIT is mostly an overfitting game and not so much and underfitting / optimization game. What was the training cost achieved by both methods?"

To confirm this, we present the training and validation set label error for the '15x8x10' network in Table 2. The label errors are reported for every second epoch in case of HF, and for every 50 epochs in case of SGD.

hf_train=[23 23.7 22.5 21.7 20.4 20.1 19.3 19.8 19.2 18.2 17.7 17.2 16.9 16.8 16.3 16.3 16 15.7 15.1 14.9 15 14.5 14.6 14.2 14.2]
hf_val=[26.4 26 25.2 24.8 24.1 23.8 23.5 23.7 23.4 22.9 22.8 22.6 22.6 22.3 22.3 22.4 22.3 22.2 22 21.8 22 21.8 22.1 21.8 22.1]
sgd_train=[26.1 18.6 14.1 12.2 9.9 8.4 7.4 6.2 5.3 4.7 4 3.4 2.8 2.7 2.2 1.8 1.5 1.3 1 1]
sgd_val=[30.4 24.3 22.2 22 21.9 22 21.9 22.2 22.2 22.1 22.4 22.1 22.4 22.4 22.4 22.3 22.5 22.7 22.4 22.6]

With weight distortion [24], the validation error does not decrease after some epochs. However, the training error keeps on decreasing as the learning is continued for both HF and SGD. Please note that HF was stopped using the stopping criteria on the validation set.

2) "... other baselines that may be more practical ... Adagrad, RMSProp, or Adam. ... useful to add some prior art in the tables for comparison."

It would have been helpful to provide a comprehensive baseline experiment. However, please note that as mentioned in [1,10], the performance of various sophisticated gradient methods including RProp, CG, and L-BFGS on training RNNs were not expected to be superior to online SGD method, especially, with the gradient clipping technique used in our paper [25, Pascanu (2013)]. For RNNs, the performance of HF method is expected to be superior to other methods since it explicitly handles the vanishing and exploding gradient problems [10]. We would like to add a few prior arts for easier comparison.

R2:
1) "... experimental comparisons with the L-BFGS baseline ..."

R1 also raises the same concern which we have tried to address in our response above. Kindly refer to those.

R3:
1) "The authors decided to train the baseline SGD to convergence (1000 iterations) even though that almost never gives the best results. They say "Note that in order to guarantee the convergence, we selected a conservative criteria compared to the reference". I can think of no other reason to do this other than to artificially handicap the baseline and to have some good-sounding results to report."

We respectfully clarify the misunderstanding about the experiments. The final performance was evaluated using the weight parameters that showed the best result for the validation set within 1000 iterations. We will modify the context for more clear reading.

2) "... no discussion of relative speed of the two methods, no convergence plots, and no objective functions reported ..."

The original CTC objective was used as the cost function throughout the paper. Regarding the convergence plots, please refer to our response to R1.

Regarding the relative speed, at least 30 CG loops were performed per iteration for HF and one CG loop requires 1.2~1.5 times computation compared to one iteration of SGD. Considering that we performed a brute force search of the learning rate and momentum parameters for SGD, similar amount of computational resources were spent for both methods.

R4:
We appreciate your encouraging comments.

R5:
1) "... (even if extremely toy) how exactly the convex approximation of CTC influences things ..."

We appreciate this insightful comment. A simple but nontrivial case is a network of 2 output neurons with a sequence of length 2. Analysis of this simple network will show how the convex approximation affects the geometry of optimization and its convergence to exact solution. These results will be included in Appendix.

2) "... any benchmark comparisons with other published methods ..."

Please refer to our response to R1 above.

R6:
1) "... what makes MDRNNs particularly difficult ... which aspects of Section 3 are new ..."

Section 3 explains the key motivation of this paper and discusses why convex approximation of CTC loss function is necessary to apply HF on MDRNNs. We will modify the context to make it clear.

2) "... the sum-of-functions optimizer ..."

We found that the sum-of-functions optimizer is quite interesting and would like to test its performance in the future. For other options, please refer to the response to R1 above.

3) "... by looking at the performance on the training data ..."

Please refer to our response to R1 for the actual numbers.

4) "... why does B map to a set and not a label sequence ..."

B maps to a label sequence. B is a surjective mapping, and S is the image of B.